# Eliciting Learner Knowledge: Enabling Focused Practice through an Open-Source Online Tool

**DOI:** 10.3390/bs12090324

**Published:** 2022-09-07

**Authors:** Meredith Thompson, Griffin Leonard, Jamie N. Mikeska, Pamela S. Lottero-Perdue, Adam V. Maltese, Giancarlo Pereira, Garron Hillaire, Rick Waldron, Rachel Slama, Justin Reich

**Affiliations:** 1Comparative Media Studies & Writing, Massachusetts Institute of Technology, Cambridge 02139, USA; 2Educational Testing Service, Princeton, NJ 08541, USA; 3Department of Physics, Astronomy & Geosciences, Towson University, Towson, MD 21252, USA; 4Center for Research on Learning and Technology, School of Education, Indiana University, Bloomington, IN 47405, USA; 5Department of Physics, Columbia University, New York, NY 10027, USA

**Keywords:** simulations, teacher education, eliciting student thinking, role play, practice-based teacher education

## Abstract

Eliciting and interpreting students’ ideas are essential skills in teaching, yet pre-service teachers (PSTs) rarely have adequate opportunities to develop these skills. In this study, we examine PSTs’ patterns of discourse and perceived learning through engaging in an interactive digital simulation called Eliciting Learner Knowledge (ELK). ELK is a seven-minute, chat-based virtual role play between a PST playing a “teacher” and a PST playing a “student” where the goal is for the teacher to find out what the student knows about a topic. ELK is designed to be a practice space where pre-service and in-service teachers can learn strategies for effectively eliciting their students’ knowledge. We review the implementation of ELK in eight teacher education courses in math or science methods at six different universities and assess (a) patterns of interaction during ELK and (b) PSTs’ perceptions of ELK and their learning from the simulation. Our findings suggest that PSTs engage in effective practices such as eliciting and probing more often than less effective practices such as evaluating and telling. Results suggest that PSTs gain experience in practicing talk moves and having empathy for students’ perspectives through using ELK.

## 1. Introduction

Eliciting and interpreting students’ ideas has been identified as one of the core skills of teaching [1], and one central aspect of ambitious and equitable STEM teaching [2,3,4,5]. Surfacing students’ knowledge is a cornerstone for best practices in teaching STEM, yet it is often overlooked by new and even experienced educators [6,7,8,9]. Research suggests that novice teachers may regard students’ conceptions as obstacles rather than pathways to learning [10,11]. As a result, PSTs need opportunities to learn the importance of and practice eliciting learner knowledge.

Practice-based teacher education programs provide PSTs with low-stakes opportunities to learn to teach. One approach is engaging PSTs in “approximations of practice” [12]. Approximations of practice purposefully forgo some of the authenticity of teaching in order to allow PSTs to practice discrete aspects of teaching, including how to elicit learners’ ideas. In typical methods courses, PSTs are introduced to eliciting learner knowledge through the study of talk moves [13] and questioning strategies [14]. PSTs practice those skills through person-to-person roleplays [15], in-class teaching exercises [16], and during their field experiences [17].

The objective of this study is to share how an open-source peer-to-peer role-playing tool can be used as an approximation of practice to help PSTs learn the complex skill of eliciting and interpreting students’ ideas as preparation for facilitating argumentation-based discussion in science and math classrooms. We start the paper by describing the construct of interest: eliciting and interpreting learner knowledge. Next, we describe current approaches for helping teachers develop that skill and we discuss some considerations when designing and using simulations for learning. We introduce the open-source online Eliciting Learner Knowledge (ELK) simulation and examine the implementation of the simulation in eight courses taught by eight different Teacher Educators (TEs). Then, we review data from the implementations to examine the types of questioning strategies and discourse patterns that PSTs used when engaging in ELK. Finally, we explore the PSTs’ perceptions of the authenticity and effectiveness of the online simulation.

### 1.1. Importance of Surfacing Students’ Conceptions

Eliciting and interpreting student thinking is the process of learning and understanding the knowledge that students have constructed about a topic [4]. Understanding what students know is essential for incorporating students’ ideas into the class [5,7]. In a classroom, the eliciting process can take many forms; teachers may ask students to write about a topic, they may engage in a full class discussion, or the teacher may have a one-on-one conversation with a student. The unifying aspect of eliciting and interpreting student thinking is surfacing students’ ideas about a domain without judgment or the intent to immediately influence those ideas [5,18,19]. Effective questioning is linked to teachers’ abilities to attend to and respond to students’ ideas at the moment [20]. Furthermore, research shows that students of teachers with a greater knowledge of students’ conceptions have stronger learning gains than students whose teachers do not [21].

### 1.2. Teacher Questioning That Promotes Student-Centered STEM Instruction

In reviewing a number of studies related to classroom discourse and formative assessment, we identified five major categories of effective elicitation of student knowledge: priming, eliciting, probing, revoicing, and you-focused questions. Effective teachers start lessons by establishing the classroom as a safe environment for engaging in a discussion about a topic [22], a strategy sometimes called framing or priming [23]. Teachers learn what their students know by eliciting their initial conceptions and ideas. Eliciting involves teachers posing open-ended questions that can help students share their initial ideas [22].

After students share their initial responses in a conversation, teachers can ask students probing questions to further understand what they are thinking [23]. Teachers who ask a series of probing questions are more effective in getting their students to explain their thinking [24,25]. Despite the benefits of probing, research suggests that PSTs are not likely to ask clarifying questions or ask their students to write down their problem-solving processes [26]. Teachers can also revoice students’ ideas into the conversation by using the student’s own words (e.g., “So what you are saying is that … Is that correct?”) [27]. In addition to helping teachers understand what their students know, revoicing demonstrates that teachers are interested in the students’ ideas regardless of whether those ideas are correct.

The words that teachers use to phrase the questions can also influence how students respond. One easy way to signal a focus on students’ ideas is to frame factual questions as you-focused questions [14]. As an example, “How would you define matter?” prompts student thinking while “What is the definition of matter?” focuses on a factual response. Open-ended questions also tend to elicit more ideas than closed-ended questions. Researchers define an open-ended question as “A question to which a number of different answers would be acceptable” (p. 864) and a closed-ended question as “A question which expects one possible response as its acceptable answer” (p. 864) [28]. In a study of three kindergarten teachers, ref. [28] found that teachers gravitated towards closed-ended questions when teaching science; however, when they did use open-ended questions, students’ responses were more detailed and used a wider range of vocabulary. We did not code specifically for open-ended questions, but we did code for closed-ended questions, as we describe below.

### 1.3. Question Types That Discourage Student-Centered STEM Instruction

There are also types of questioning patterns that are not conducive to student-centered instruction. For example, closed-ended questions do not encourage students to be generative in their responses. Researchers [28] define a closed-ended question as: “A question which expects one possible response as its acceptable answer” (p. 864); this is juxtaposed with an open-ended question, defined as: “A question to which a number of different answers would be acceptable” (p. 864). Of particular focus in our study is a type of close-ended question: yes/no questions. In mathematical problem solving, asking a series of closed-ended questions can oversimplify the problem for the student [29]. Yes/no questions are not helpful in getting students to describe their thinking in detail, as students often will simply answer ‘yes’ or ‘no’ without giving any additional information” [30].

Another technique to avoid in the eliciting phase is telling the student the right answer. For example, telling a student how to solve a math problem interrupts the child from their own thinking process [29] and can move them from an active participant in problem solving to a passive observer. Teachers may unwittingly engage in telling with a series of questions that lead students towards a specific answer, a technique called funneling [20].

Another strategy that discourages students’ ideas is an overemphasis on evaluating students’ responses. Constant feedback from the teacher shifts the frame away from the students’ ideas and towards the traditional teacher Initiates, student Responds, teacher provides Feedback (IRF) format, where providing feedback and evaluating are analogous [31]. Furthermore, telling and evaluating patterns suggest the teacher is the source of knowledge and the primary controller of what happens in the classroom [32].

### 1.4. Practicing Eliciting and Probing Learner Knowledge through Simulations

One way to learn and practice complex skills is by engaging in simulations. A simulation is a way of modeling or approximating a real situation as a learning environment to gain new skills [33]. A meta-analysis of 146 studies of simulations in higher education [33] notes that simulations are effective learning tools because they give “the opportunity to alter and adjust some aspects of reality in a way that facilitates learning and practicing (e.g., they address less frequent events, shorten response time, provide immediate feedback to the learner, etc.)” (p. 502). Successful simulations also have defined outcomes, clearly articulated learning objectives, and ways to assess whether those learning objectives have been met with both performance measures and self-reporting measures [34,35].

The design of simulation-based learning experiences for PSTs in this study is informed by the instructional design model: Ten Steps to Complex Learning [36]. This holistic instructional design model emphasizes that complex skills—such as teaching—are characterized by the diversity of discrete elements in task performance and the high degree of coordination required for high-quality performance. At any given moment in a lesson, teachers are recalling key features of the lesson content, facilitating activities, monitoring the clock, and observing student work and behavior for evidence of comprehension. These simultaneous activities put a high cognitive load on teachers [37]. For novices, it is challenging to both perform this entire complex assemblage and work on improving discrete parts at the same time. This observation is commonplace in many learning environments: It is why music teachers and athletic coaches employ drills alongside recitals and scrimmages. According to the model, these drills or discrete practice opportunities serve as “part-task practice”, where novices can develop automaticity in managing recurrent features of the complex assemblage of the whole. [38] refers to the part-task practice of discrete teaching elements as “teaching drills”. Since effective drilling involves high levels of repetition, digital simulations may play an important role in providing PSTs with sufficient practice time and repetition to improve in discrete elements that can be combined into the complex assemblage of teaching.

One measure of the efficacy of the simulation is the degree to which users—in our case, PSTs—perceive the simulation as authentic. While simulations can reduce the complexity of teaching, the experience must feel real enough for the PSTs to be able to activate the correct skills that they can transfer into actual classroom interactions. One measure of the authenticity of a classroom-based simulation is the extent to which the PST perceives the content and objectives of the discussion and associated background information to be similar to what would occur in the real classroom.

### 1.5. Role Plays

Many simulations incorporate some form of role-playing. [39] describe role play as a “social or human activity in which participants ‘take on’ or ‘act out’ specific ‘roles’, often within a predefined social framework or situational blueprint” (p. 155). Next, we will define three types of role plays that are used by PSTs, and discuss some of the challenges of each type.

One type of simulation often used in teacher education is in-person peer-to-peer role play. PSTs in the same class can practice these skills by taking on the role of a teacher or a student and engaging in a conversation with each other. Role plays allow PSTs to explore pedagogical strategies, enable TE feedback, and increase PSTs’ confidence in their teaching skills [15]. Role plays can be helpful in (a) practicing effective communication, including listening to and attending to their partners’ ideas [40], (b) building empathy for others [41], and (c) reflecting on their own role in the conversation.

Another strategy for practicing eliciting student ideas is through clinical role-playing simulations. Clinical simulations are a form of role-playing that involve an external or specially trained actor or actors. Borrowing from medical education, [42] incorporated clinical simulations into their teacher preparation program. Clinical simulations have been used to introduce PSTs to challenging situations such as dealing with angry parents [43,44], responding to difficult discussions about race in the classroom [45], and learning how to interact with students during one-on-one conversations [26].

Ref. [26] developed clinical simulations for assessing PST’s ability to elicit learner knowledge. In these simulations, a PST engages in a role play with a trained expert who role plays a standardized student who has completed a mathematics problem [18]. PSTs demonstrate their proficiency in eliciting knowledge by interacting with the student. In comparing field interviews with the simulation, they found that the simulation is a viable method for having PSTs demonstrate their skills [18].

Another avenue for interaction is through digital clinical simulations. Digital clinical simulations include technological interfaces, allowing remote actors to engage in the simulations and enabling one actor to play more than one student [46,47]. Similar to in-person clinical simulations, a digital simulation experience may become more authentic than peer-to-peer role plays as the actor is more familiar with the students they are portraying.

### 1.6. Challenges of Role Plays

In addition to the benefits of role-playing, there are also challenges for both peer-to-peer and clinical simulations. During in-class role plays, participants may emphasize entertaining their peers above achieving the learning objective of the role play [48,49]. Furthermore, in-person role plays lack the benefit of an easily recorded artifact for review and reflection, something that is possible with digital simulations [50]. While it would be possible to audio record the conversations, audio recording individual conversations in a noisy classroom does not result in high-quality recordings. In a role play, participants need to have enough information to make the conversation happen, but not so much that the players are overwhelmed [40]. Preparing PSTs to accurately embody their role in the activity takes time and resources and is important in developing a level of authenticity in the simulation. In-person role plays take careful planning and classroom management and may not create an easy artifact for reflection.

With in-person clinical simulations, the standardization of the student and selection of the problem (all PSTs see the same problem) is less authentic than interacting with actual students but has the benefits of being able to focus on one specific skill and allows a more systematic assessment across a class of PSTs [51]. One-on-one clinical simulations that are either in-person or virtual require PSTs to allocate additional time outside of class, require additional time for training the actor engaged in the simulation, and must include the cost of paying the actor. Both peer-to-peer and clinical simulations have advantages and challenges; next, we will share a potential solution to some of these challenges through a simulation called ELK.

### 1.7. Eliciting Learner Knowledge (ELK)

ELK is a text-based chat where one player role plays a student while the other player role plays a teacher. The PST in the student role is given a student profile detailing background knowledge about a given concept. The PST in the teacher role is given a learning objective for the class. Both teacher and student are provided with contexts such as the problem (in math) or investigation (in science) the class previously completed, and the overall learning objective for the unit. During the enactment of ELK, players log onto the system with a username and password, select whether they are role-playing a teacher or a student, find the appropriate simulation from a list, find their assigned partner, review their profiles and then start the chat. It is then the job of the PST in the teacher role to elicit the student’s knowledge. This is done through a seven-minute interaction between the PSTs via text chat messaging. The choices of texting and seven minutes were the result of extensive simulation testing [52]. Seven minutes ensures a level of challenge by giving the players enough time for a conversation but not unlimited time, as teachers do not have unlimited time in the classroom. Texting rather than talking is preferred for the ability to capture an immediate artifact in the form of a transcript. After the 7 min, the PSTs fill out a short true/false quiz to see how well the teacher was able to learn what the student knows (from the PST playing the student) and to portray what the student understands (according to the PST playing the student). A summary of the ELK simulation topics appears in Table 1. Note that some students have accurate conceptions, and some do not. A screenshot of how the ELK simulation would appear for the PST role-playing the teacher appears in Figure 1. Screenshots of the entire simulation appear in the Appendix A.

We created ELK to be a tool for TEs to help PSTs learn and practice questioning skills. This study focuses on whether ELK is a viable simulation for incorporation into teacher education courses. The research questions that guide this paper are as follows:What types of questioning strategies do PSTs employ during the ELK simulation?How do PSTs perceive the goal(s) and authenticity of the ELK simulation and what have they learned from participating in the ELK simulation?

## 2. Materials and Methods

### 2.1. Study Context

We partnered with four elementary method TEs from three universities in the spring of 2021 and four secondary method TEs from three additional universities in the fall of 2021. Each TE integrated the ELK simulation in one section of their mathematics or science methods course. This was part of a larger project where teacher educators integrated a series of teaching simulations in their methods class, starting with ELK and then proceeding to mixed-reality simulations and a new virtual reality classroom simulator [53]. The key characteristics of the courses at each teacher education site are summarized in Table 2.

Each TE reviewed questioning strategies with their PSTs as a way to prepare them for the ELK simulation. Seven of the eight TEs had their PSTs engage in ELK during class time. One TE assigned ELK for homework; PSTs in that class had to set a time when they would meet and do ELK together. The lead researcher was present during class time through a videoconference call to assist the class with logging onto the ELK platform. Each of the TEs used slides with specific instructions about the activity that were provided by our research team, with slight modifications. All of the TEs had their PSTs do at least two rounds of ELK so each PST had a chance to play the role of a student and a teacher. In their debrief after ELK, all of the TEs had the PSTs reflect on and analyze the transcript generated from the ELK text-based conversation. Each TE did ELK with their class at least twice; four TEs had their PSTs engage with ELK an additional two times such that the PSTs each played a teacher and a student two times.

### 2.2. Transcript Data (RQ1)

Each pair of PSTs generated a transcript from their ELK discussion, which was saved in the ELK platform. Conversations were downloaded from the platform as a .csv file, with each line of the conversation captured in one row of the data file. Each row also included the participant’s username, a timestamp, and a conversation ID that linked the two partners together. The .csv was saved as a database and a set of columns was created for each of the codes in the coding system described in the previous teacher questioning sections (Section 1.2 and Section 1.3), exemplified in Table 3 below.

### 2.3. Survey Data (RQ2)

After all of the debrief and reflection activities on ELK were completed, PSTs who were participating in the research study completed an online survey with questions about argumentation-based discussions and their perceptions of what they learned from the ELK simulation. The three open-ended questions included in this analysis were the following: 1. What did you learn from role-playing a teacher in the ELK activity? 2. What did you learn from role-playing a student in the ELK activity? 3. In your own words, what was the goal for the discussion you facilitated in the ELK session?

### 2.4. Data Analytic Methods

We used a mixed-methods approach for the study. This involved qualitative coding of conversation utterances with predetermined codes, quantitative summaries of the utterances, and both qualitative and quantitative analyses of PSTs’ survey responses. A subset of the authors used extant research to form a coding system that we used to track positive and negative questioning strategies and discourse patterns [52]. The coding list appears in Table 3.

To determine the pattern of PST responses to the simulation (RQ1), we calculated descriptive statistics (frequencies and standard deviations) on the coding data generated from all teacher utterances. We did not analyze the “student” conversation lines for this study, only the “teacher” utterances were coded. Each utterance could be coded with more than one code. To answer RQ2, we calculated descriptive statistics (frequencies) for each of the survey questions that used four-point Likert Scales (forced-choice) options. Two researchers reviewed the open-ended questions for common themes within the questions by independently reading the responses and employing thematic analysis [54]. Next, the researchers shared the themes, created a coding system, and independently coded the open-ended responses. The researchers then reviewed each coded segment and discussed discrepancies until there was 100% agreement on how to code each response [55].

### 2.5. Patterns of Questioning Strategies (RQ1)

All ELK conversations were downloaded as a spreadsheet where each row in the spreadsheet corresponded to the response a PST typed before hitting “send”, which could be a single word or multiple sentences. Two raters coded the conversations. To avoid the kappa paradox, we used Gwet’s AC to gauge the reliability of coding [56]. Similar to Cohen’s kappa, Gwet’s AC is scored on a scale between 0 and 1; values above 0.7 are generally considered acceptable [57]. Our results suggest that the coding was extremely reliable, with values of 0.998 (telling), 0.999 (priming), and 1.00 for all other codes (See Appendix B).

### 2.6. PST Perceptions of ELK Analytic Approach (RQ2)

The PSTs’ perceptions of ELK were drawn from the PST task survey, which included questions about their preparation for ELK, their perceived goals for ELK, whether the goals were met, their perceived learning through ELK, and the authenticity of ELK. For this paper, we reviewed all survey data and extracted a subset of four questions to analyze: two open-ended questions and two Likert scale questions.

Two raters selected a subset of open-ended questions—what they perceived as the goal of ELK, what they learned from role-playing a teacher, and what they learned from role-playing a student—from the survey that addressed the ELK simulation. Each rater then provided categories to the other rater and both coded questions independently. After this, percentage agreement was calculated and the two researchers discussed and resolved any differences in coding.

The two Likert scale questions both related to the PSTs’ perceptions of task authenticity. One asked how authentic the simulations were when compared to a normal classroom with actual students. Possible ratings were: very authentic, somewhat authentic, minimally authentic, and not authentic. The other asked for each PST’s level of agreement—agree, somewhat agree, somewhat disagree, or disagree—with the following statement: “My performance accurately reflects my ability to facilitate classroom discussions with real students”. We calculated frequencies for each of the ratings for each of these questions.

### 2.7. Sample

The sample consisted of 57 PSTs from six universities; 43 identified as female, 11 identified as male, and 3 participants did not share their gender. PSTs’ self-reported race was white (43), Asian (6), Black (3), American Indian (1), and 4 did not provide their race. Three PSTs listed their ethnicity as Latinx; two of those three selected white for race and Latinx for ethnicity. In total, 56 l of the 57 students completed all of the survey questions. These participants were selected because they were PSTs currently enrolled in an elementary or secondary teacher education program and were in a course with one of the participating TEs from one of the six universities that participated in the study.

Next, we provide information on the entire dataset including the length, number of teacher lines per conversation, and the total number of conversations. As shown in Table 4, we analyzed a total of 95 conversations that had an average length of 15.3 lines. The number of lines that were uttered by the PST role-playing the “teacher” was about half of the conversations on average, suggesting that there was a balance in the exchange between the PST role-playing the teacher and the PST role-playing the student. The PSTs engaged in multiple ELK conversations, however, only conversations where both PSTs consented to be in the study were included in the analysis.

## 3. Results

RQ1. What types of questioning strategies do PSTs employ during the ELK simulation?

In order to gain an understanding of the structure and content of these conversations, we first present summary statistics for the entire dataset and then evaluate two specific conversations to demonstrate how we used the codes to assess the effectiveness of the exchange. Since conversation length varied between different role-playing partners, we report each of the following frequencies of appearance of codes standardized by conversation.

In Table 5, we share the mean of each type of questioning strategy in the coding scheme. The mean refers to how many of these types of questions, on average, appeared in the PSTs’ conversations. Results show that, on average, there were 0.32 *priming* statements per conversation across both elementary and secondary PSTs; thus, not every PST used priming during their ELK round. There were 1.45 *eliciting* statements per conversation, meaning that, on average, there was at least one eliciting question during the conversation, and some PSTs had more than one. The most frequently occurring strategy was *probing*, at 4.88 statements per conversation on average. *Revoicing* occurred 0.55 times and *you-focused questions* 4.49 times per conversation on average.

Overall, PSTs used more productive strategies than counterproductive strategies. While instances of telling and funneling were rare (0.2 times per conversation, on average each), there were more examples of evaluating (1.16 times per conversation), which is more than the PSTs engaged in the productive strategies of priming or revoicing. Table 6 and Table 7 show two examples of conversations, along with the codes applied to these examples. The first conversation demonstrates a PST, “Teacher A,” who employs a number of productive questioning strategies; the second conversation shows another PST, “Teacher B,” who uses a number of counterproductive questioning strategies.

Teacher A was effective at using eliciting and probing questions while avoiding evaluating and telling, which is the main goal of ELK. The conversation also included you-focused questions, which focuses the conversation on the student’s ideas.

Teacher B used eliciting and probing strategies in this conversation, but also was evaluating the responses of the PST role playing the student and engaged in telling when the teacher drew the conclusion in the final line. Some of the questions are you-focused, but not all of them.

RQ2. How do PSTs perceive the goal(s) and authenticity of the ELK simulation and what they learned from participating in the ELK simulation?

PSTs were asked for their perceptions of the goal of ELK in an open-ended question. We found four main themes in PSTs’ perceptions of their goals of ELK. The 57 PSTs identified the goal of ELK to be: 1. having the teacher practice questioning, 2. understanding the student’s thought processes, 3. having the student explain their thinking, and/or 4. guiding the student towards a specific conclusion. Results are shown in Table 8. PSTs identified more than one goal in their response, so the overall percentage does not add up to 100%.

The first three goals identified by PSTs in Table 8 PSTs are aligned to the intended goals of ELK. It is noteworthy, though, that 9 of 58 PSTs (16%) also perceived that teaching the student or guiding the student to the correct answer was a goal of ELK. These responses did not align with the intention of ELK as a way to elicit, and not change, the student’s ideas. For example, one explained that “I wanted to get the student to see why their strategy was good in some cases but not all”. Another mentioned, “lead[ing] the students to the proper conclusion without giving them the conclusion straight up”. This suggests that some PSTs’ ideas about understanding what their students know are focused squarely on correcting creative conceptions, rather than fully understanding what those conceptions may be.

### 3.1. Learning from Playing the Role of the Teacher

Learning from role-playing a teacher was associated with two main themes: learning the importance of how to elicit learner knowledge and learning that it is challenging to accomplish. Twenty-three (41%) of PSTs noted they learned the importance of questioning strategies, asking thoughtful questions, predicting what students know, and asking probing, open-ended questions. One student wrote that “I learned that asking thoughtful questions can really make understanding of the content seem easier”. Eleven PSTs (20%) mentioned that ELK made them realize how difficult it was to elicit learner knowledge. Some mentioned difficulty in asking open-ended questions, in following up when students did not answer the question they were asked, when students sometimes had surprising responses, and in managing the challenge of the limited timeframe of 7 min. One student explained that “I learned that it was much harder to ask questions to students that promote thinking on the spot”. There were some less common themes, including understanding how they needed a clear plan (6 PSTs, 11%) and recognizing the importance of time management (4 PSTs, 7%).

### 3.2. Learning from Playing the Role of the Student

PSTs also were asked what they learned from playing the role of a student. Twenty-one PSTs (37%) mentioned that role-playing a student helped them gain empathy for the student’s perspective. PSTs empathized with students: feeling potentially judged in the conversations and that it is important for teachers to not “seem like you are judging”; not knowing how to articulate their thinking (“it was difficult to get your ideas across”) or being nervous when being questioned by the teacher, and that “the teacher’s responses contribute to this anxiety”. PSTs learned that it can be challenging to encourage students to talk (18%), that the task was difficult (12.5%), that asking good questions is important (11%), and that students have different ways of solving problems (11%). One PST explained, “Every student grasps knowledge differently, and we must accommodate that”.

### 3.3. Authenticity of ELK

In total, 15 PSTs (27%) rated ELK as very authentic, 25 (44%) rated ELK as somewhat authentic, 13 (23.3%) rated ELK as minimally authentic, and only 2 (3.6%) said it was not authentic. Thus, over two-thirds of participants felt that the ELK simulation was very or somewhat authentic. Relatedly, about half of PSTs (51%) disagreed or somewhat disagreed (and 49% agreed or somewhat agreed) with the statement that ELK was an accurate reflection of their performance in engaging in a discussion with a student. In a follow-up question, PSTs mentioned that it would be different to interact with real children in a face-to-face setting. PSTs mentioned that “it’s different when interacting with real students” and that having the student read off a script “felt robotic”. While the PSTs had suggestions of answers to give (for the student), they did not have the script for an entire conversation. PSTs also noted that ELK was an unfamiliar format, but that they found the interface was easy to use.

## 4. Discussion

RQ1. What types of questioning strategies do PSTs employ during the ELK simulation?

Our results align with [26]’s findings that most PSTs exhibited both productive and counterproductive strategies in ELK. The PSTs who participated in ELK engaged in the productive strategies of *eliciting, probing*, and focusing on student thinking during the conversations, similar to the PSTs in [25]. The most common productive talk move was *probing*; PSTs asked more *probing* than *eliciting* questions. In fact, similar to PSTs in [30] and [18], PSTs spent most of the conversation *probing* for students’ ideas. PSTs used you-focused questions that zeroed in on their peers’ (who played the students) ideas rather than asking for scientific facts, as recommended in [14]. We also saw some behaviors that were not supportive of eliciting learner knowledge. In line with the findings in [28], the most common counterproductive strategy that PSTs had was asking closed-ended questions, especially *yes/no questions*. Closed-ended questions are not as helpful as open-ended questions in eliciting students’ responses; a series of closed-ended questions can become a way of *funneling* students to a specific answer [20]. *Probing* questions can also be *yes/no questions*, however, in the context of a short conversation, efficiency in question asking is important, so asking one open-ended question is more efficient than a closed-ended question that requires a follow-up. Some PSTs evaluated students’ responses, providing them feedback about whether their responses were on the right track. Often PSTs considered this to be a way of encouraging the student, however, in reflecting on the activity, PSTs recognized that this could be a way of shifting the conversation away from the students’ ideas toward the “right answer”. While not as common as closed-ended questions and *evaluating*, PSTs also did their share of *telling* in the conversations, especially in the form of *funneling* students to a specific answer [20]. Similar to [18], the design of the ELK profile prompted PSTs to engage in questioning strategies about a specific science or math topic through peer-to-peer role play. The transcripts of ELK become a means to assess PST knowledge and help PSTs identify strengths and weaknesses in their teaching skills.

RQ2: How do PSTs perceive the goal(s) and authenticity of the ELK simulation and what they learned from participating in the ELK simulation?

Survey results indicate that PSTs understood the goals of the ELK activity as learning and practicing questioning strategies, understanding students’ thought processes, and getting students to explain their thinking. Engaging in ELK allowed PSTs to practice talk moves and questioning strategies during a conversation, effectively putting the “vocabulary” of talk moves—specifically those that aimed to elicit student thinking—into the context of a conversation [40]. PSTs also felt ELK helped them practice gathering, understanding, and attending to students’ ideas, making these teaching moves a part of their teaching practice early in their teaching career as recommended in [6].

Similar to [41], PSTs reported that they gleaned various benefits from playing the student role. PSTs developed empathy for a student’s perspective through gameplay; playing ELK as a student reminded the PSTs not to appear judgmental of the student’s ideas, and how it can be challenging for students to describe their thoughts. There are certainly benefits from having an expert play the role of a student (e.g., [6,26]), but there are also benefits to well-scaffolded peer-to-peer role play. Furthermore, having PSTs play both the teacher and the student role shifts the focus from a teacher-focused narrative to one that also includes the student.

Any simulation sacrifices some degree of authenticity in order to reduce complexity [12]. The key for simulations is to have enough authenticity for the task to be considered beneficial practice [58]. The results from the PST survey indicate that a majority of the PSTs believe ELK is authentic, and that they felt the activity met the goals of practicing eliciting learner knowledge.

A common theme among the PSTs was the feeling that they would have performed better in an actual conversation with a student. We do not have data from this study to refute this statement. However, we do know that conversations with real students have higher stakes and are more complex than simulations. Current data suggest that most PSTs understand the goals of the simulation and feel that they can practice working towards those goals during ELK. The study results suggest that ELK is authentic enough to provide PSTs with practice in eliciting a one-on-one conversation. Future studies will look at PSTs performance over multiple instances of ELK and also simulations of different complexity to see whether PSTs exhibit more positive questioning practices. Future research will also see whether the skills that are learned are transferable to actual conversations.

## 5. Conclusions

This study investigated elementary and secondary PST’s performance and PST’s perceptions of learning during the ELK simulation. During the conversations, PSTs exhibited some positive strategies for ELK in *eliciting*, *probing*, and selecting *you-focused questions* over fact-focused questions. They also exhibited some negative strategies of *evaluating* and *telling* and an emphasis on lower-order questioning through closed-ended questions. In role-playing a teacher, PSTs reported that they learned how to elicit learner knowledge and that eliciting is a challenging skill. Role-playing a student enabled the PSTs to gain empathy for the student’s perspective in a one-on-one conversation. Furthermore, a majority of the PSTs perceived the activity as authentic.

In the future, we would like to incorporate more automated responses into the platform to provide even more just-in-time feedback to players. All of the PSTs analyzed their own transcripts as part of the debrief of the activity, which supported PSTs’ understanding of productive strategies. We envision automated feedback for participants that could occur during the conversation so the PST could change their tactics even within the conversation. We are also working towards having a large enough data bank of responses so that PSTs could have a conversation with a chat bot, eliminating the need for a peer altogether. Such a setup would allow for more flexibility for the player, but a downside to consider is that it eliminates the learning experience potential of the PST playing a student in the simulation. Future research will also compare PSTs responses with additional rounds of ELK, conversations in more complex simulations, and eventually, following PSTs into the classroom to see if the impact of ELK is traceable to classroom conversations. This study reinforces the idea that ELK is an open-source simulation with sufficient authenticity of action to enable PSTs to learn and practice how to have effective student-centered conversations in low-stakes settings to prepare for high-stakes interactions with real students.

## Figures and Tables

**Figure 1 behavsci-12-00324-f001:**
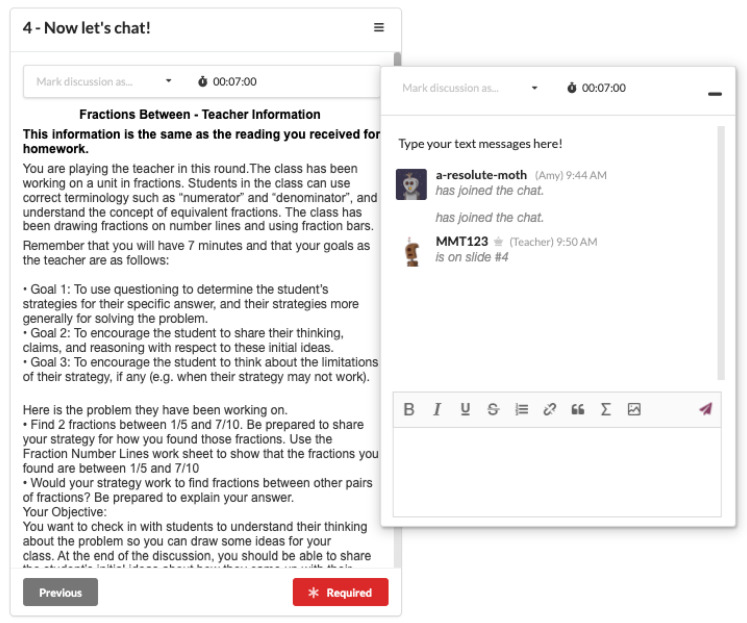
Screenshot of ELK showing profile information on the left and chat box on the right for a PST playing the “teacher” role.

**Table 1 behavsci-12-00324-t001:** Summary of ELK simulation topics.

Title	Level	Primary Question	Student 1	Student 2
Conservation of Matter	Elementary Science	Is matter conserved when paper is crumpled or water is frozen into ice?	Charlie: Thinks matter increases because water volume increases upon freezing.	Dana: Thinks matter is conserved because weight does not change.
Finding Fractions Between	Elementary Math	How can we find a fraction between two numbers?	Amy: Finds the least common denominator.	Scott: Finds numbers between the two numerators and between the two denominators.
Keeping the Heat	Secondary Science	Which cup did the best job of keeping the hot chocolate hot (foam or paper)?	Victor: Foam cup is best because does not let the cold into the cup (cold as a substance).	Rosa: Foam cup is best because the heat bounces off the walls better in the foam cup than in the paper cup.
Rate of Strawberry Picking	Secondary Math	How can we calculate the rate of strawberry picking given a table of values?	Braden: Calculates rate using proportional relationships (does not show steps).	Emilie: Calculates rate using a table.

**Table 2 behavsci-12-00324-t002:** Key Characteristics of course for each teacher education site.

Semester	TE	University Site	Content Area	Methods Course Type	Credit Hours	Course Format
Spring 2021	TESci1	U1	Science	Elementary	3 credits	Synchronous, online
	TESci2	U2	Science	Elementary	3 credits	Synchronous, online
	TEMat1	U3	Mathematics	Elementary	1 credit	Synchronous, online
	TEMat2	U1	Mathematics	Elementary	2 credits	Hybrid, online
Fall 2021	TSSci3	U1	Science	Secondary	3 credits	In-person
	TSSci4	U4	Science	Secondary	3 credits	In-person
	TSMat3	U5	Mathematics	Secondary	4 credits	In-person
	TSMat4	U6	Mathematics	Secondary	3 credits	In-person

**Table 3 behavsci-12-00324-t003:** Productive and unproductive strategies for eliciting learner knowledge.

Strategy	Example Teacher Question or Other Prompt
Productive strategies for eliciting learner knowledge:	
Priming	“Let’s discuss the freezing water demonstration.”
Eliciting	“What did you observe when we froze the water in the plastic bottle?“
Probing	“Why do you think the volume of the water increased?”
Revoicing	(After a student says, “Something must be happening with the water molecules.”) “So what you’re saying is that something is happening with the water molecules? What could be happening?”
You-focused questions	“What do you think heat is?”
Counterproductive strategies for eliciting learner knowledge:	
Evaluating	“Great! The water volume does increase.“
Telling	“Water volume increases when the water becomes ice.”
Funneling	(After a student says, “I think the volume of the water increased but the mass stayed the same.”) “So are you saying that matter was conserved?”
Yes/No questions	“Did you add or take away any paper when you crumpled it up?”

**Table 4 behavsci-12-00324-t004:** Descriptive statistics about ELK conversations.

Statistic	Elementary Only	Secondary Only	Elementary + Secondary
Number of participants	26	31	57
Number of conversations	59	35	94
Mean number of lines for the entire conversation length	14.4	17.2	15.3
Number of lines for the entire conversation length standard deviation	6.64	4.63	6.10
Mean number of teacher lines	7.5	8.4	7.8
Number of teacher lines standard deviation	3.49	2.14	3.07

**Table 5 behavsci-12-00324-t005:** Mean frequency of questioning strategy codes by conversation.

Questioning Strategy	Elementary Mean (Standard Deviation)	Secondary Mean (Standard Deviation)	Mean − Elem + Secondary (Standard Deviation)
Productive strategies for questioning			
Priming	0.39 (0.64)	0.2 (0.58)	0.32 (0.62)
Eliciting	1.56 (0.67)	1.29 (0.51)	1.45 (0.63)
Probing	4.71 (2.54)	5.26 (1.86)	4.88 (2.33)
Revoicing	0.56 (0.67)	0.54 (0.87)	0.55 (0.75)
You-Focused Questions	4.40 (2.48)	4.63 (1.93)	4.49 (2.30)
Counterproductive strategies for questioning			
Evaluating	0.88 (1.18)	1.66 (1.72)	1.16 (1.45)
Telling	0.25 (0.77)	0.11(0.32)	0.2 (0.64)
Funneling	0.25 (0.54)	0.11(0.32)	0.2 (0.47)
Yes/no question	2.17 (1.60)	1.0 (1.2)	1.7 (1.6)

Note: Many utterances had more than one code applied. For example, many of the probing questions were classified as revoicing.

**Table 6 behavsci-12-00324-t006:** Example conversation with more productive than counterproductive questioning strategies.

Role	Line	Codes
Teacher A	Can you tell me what you remember from our changing paper and freezing water investigations?	eliciting, you-focused
Student	I remember the water we froze into ice. The water and ice weighed the same.	
Student	The paper was flat, crumpled, and ripped.	
Teacher A	I remember that too. Based off that information, what can you tell me about the amount of matter from the beginning of the experiment to the end?	probing, you-focused,
Student	The matter stayed the same because the weight stayed the same from beginning to end in both experiments.	
Teacher A	What evidence do you have to support your claim that matter was conserved in both investigations?	probing, you-focused,
Student	In the paper experiment, the weight was 4.6 g when it started flat. When it was crumpled it was still 4.6 g. When it was ripped into tiny pieces it was still 4.6 g.	
Teacher A	I like your supporting evidence. Do you believe that matter is conserved in things besides paper and water?	evaluating, probing, yes/no question, you-focused
Student	Yes.	

**Table 7 behavsci-12-00324-t007:** Example conversation with more counterproductive than productive questioning strategies.

Role	Line	Codes
Teacher B	Hi Charlie, what do you think about the paper? Do you think we gained or lost matter?	eliciting, you-focused
Student	I think when we crumpled up the paper we got more.	
Teacher B	What makes you think that?	probing, you-focused
Student	The crumpled up paper takes up more space than the flat paper.	
Teacher B	Okay, good observation. Did you add or take away any paper when you crumpled it up?	probing, evaluating, yes/no question, you-focused
Student	No, I did not	
Teacher B	So, for the paper, if nothing was added, do you think changing the shape also changes the amount of matter we have?	probing, you-focused
Student	No, it will not	
Teacher B	Perfect! Now same for the water, did we add or take any water out of the bottle when we froze it?	eliciting, evaluating
Student	No, the cap stayed on!	
Teacher B	Exactly! So the same can be said for the water, just because we changed the shape or state of it, we still have the same amount of matter.	evaluating, telling

**Table 8 behavsci-12-00324-t008:** PSTs’ perceptions of the goals of ELK from open-ended survey questions.

Categories	Percentage Mentioned by PSTs (N = 57)	Sample Quote
Have the teacher practice questioning	43%	The goal was to get the teachers to practice engaging in student-led conversations that would allow them to explain and elaborate on their work.
Understand student thought processes	41%	To elicit student responses that encouraged them to prove their claim with reasoning and support. To have students explain their thoughts.
Have students explain their thinking	38%	To get my students to explain their work and justify their steps.
Guide to specific conclusion (e.g., teach/explain)	16%	The goals were to lead the student in a discussion and hopefully help them to draw the correct conclusion.

## Data Availability

Additional materials and data from this study can be found that the following website: https://osf.io/378bx/files/osfstorage.

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
