# Peer review of "Eliciting Learner Knowledge: Enabling Focused Practice through an Open-Source Online Tool"

_behavsci, 2022, doi:10.3390/bs12090324_

Round 1

Reviewer 1 Report

1. Please mention the reason of choosing ELK instead of other Simulation methods ?

2. Please explain the participants selection criteria!

3. In the conclusion parts, please explain the results as answers for the research questions!

4. the introduction is too long, make it more concise

Author Response

General Feedback from both of the Reviewers (included in both responses) 

G1. Is the content succinctly described and contextualized with respect to previous and present theoretical background and empirical research (if applicable) on the topic? (can be improved, Reviewer 1 / must be improved Reviewer 2) 

Response: We have added a theory of learning to support the work and have significantly streamlined the introduction to make the introduction more succinct.  The remaining sections are important for framing our argument and providing a sufficient background to the reader. 

G2. Are all the cited references relevant to the research? (yes, Reviewer 1/ can be improved, Reviewer 2) 

Response: We have removed some of the references that are tangentially related to the research. We have reviewed the rest of the references and believe they are relevant and important to include in the paper.  

G3. Are the research design, questions, hypotheses and methods clearly stated? (can be improved, Reviewer 1/ can be improved, Reviewer 2) 

Response: We have reviewed the research design, questions, and methods and have made some edits and changes to improve the clarity of the section. This section only has limited changes, as we felt the information was succinct and essential to understanding the study. 

G4. Are the arguments and discussion of findings coherent, balanced and compelling? (yes, Reviewer 1/ must be improved, Reviewer 2) 

Response: We have revised the discussion to include more links to the background section and have answered the research questions in the conclusion to be more coherent. 

G5: For empirical research, are the results clearly presented? 

Response: We have reviewed the results and have streamlined and clarified the results section. Specifically, we have revised the description of the data in Table 5 to draw directly from the table, and have reworked sections 3, 3.2 and 3.3. We have removed the figure and focused on the final question in the figure by placing it in the text. 

G6: Is the article adequately referenced? 

We have reviewed the references in the document, and believe we have supported our arguments with specific, pertinent references.  

G7: Are the conclusions thoroughly supported by the results presented in the article or referenced in secondary literature? 

We have revised the discussion to include more direct links to the background research mentioned in the paper, and have revised the conclusion to clearly answer the research questions.  

Specific reviewer responses 

R1. 1. Please mention the reason of choosing ELK instead of other Simulation methods? 

We have included the following explanation for choosing ELK.  

We created ELK to be a tool for TEs to help PSTs learn and practice questioning skills. This study focuses on whether ELK is a viable simulation for incorporating into teacher education courses. 

R1.2. Please explain the participants selection criteria! 

We have added the following:  

“These participants were selected because they are PSTs currently enrolled in an elementary or secondary teacher education program that opted into the study and that are enrolled in the classes of our partner TEs at our partner universities.” 

R1.3. In the conclusion parts, please explain the results as answers for the research questions! 

In the conclusion, the response for research question 1 is in the first and second sentences. We added sentences that summarized PSTs’ perceptions of learning (sentence 3 and 4) and their perceptions of the authenticity of the task.  

“This study investigated elementary and secondary PST’s performance and PST’s perceptions of learning during the ELK simulation. During the conversations, PSTs exhibited some positive strategies for ELK in eliciting, probing, and selecting you-focused questions over fact-focused questions. They also exhibited some negative strategies of evaluating and telling and an emphasis on lower-order questioning through closed-ended questions. In role playing a teacher, PSTs reported that they learned how to elicit learner knowledge and that eliciting is a challenging skill. Role playing a student enabled the PSTs to gain empathy for the student’s perspective in a one-on-one conversation. A majority of the PSTs perceived the activity as authentic, easy to use, and felt they would improve their performance with additional practice.” 

R1.4. the introduction is too long, make it more concise 

We have reviewed the introduction and significantly reworked it to make it more focused on eliciting learner knowledge.  

Reviewer 2 Report

Review Report

Article title: Eliciting Learner Knowledge: Enabling Focused Practice through an Open-Source Online Tool

This study examined the patterns of discourse and perceived learning of preservice teachers (PSTs) through the participation in an interactive digital simulation called Eliciting Learner Knowledge (ELK). The authors of this study asked themselves the following research questions:

1.     What types of questioning strategies do PSTs employ during the ELK simulation?

2.     How do PSTs perceive the goal(s) and authenticity of the ELK simulation and what they learned from participating in the ELK simulation?

The authors made a significant effort to guide the reader through the overall research study. The problems of this study are detailed below.

Specific comments:

11. The introduction of the manuscript should contain a clearly formulated objective of the research.

22. The paper would be significantly improved with the addition of more details about educational learning theories which the authors rely on.

33. The discussion could be enlarged by explaining whether any similarities and discrepancies with other published data have been identified and accounted for.

Author Response

Reviewer 2 - Specific reviewer responses: 

R2.1 The introduction of the manuscript should contain a clearly formulated objective of the research. 

We have inserted the following research objective into the manuscript. It is the first sentence in the final paragraph of the 1. Introduction section. 

The objective of this study is to  we share how an open-source peer-to-peer role-playing tool can be used as part of a set of learning activities and resources to help PSTs learn the complex skill of eliciting and interpreting students’ ideas as a preparation for facilitating argumentation-based discussion in science and math classrooms. 

R2.2 The paper would be significantly improved with the addition of more details about educational learning theories which the authors rely on. 

We have added a section on the Theory of Complex Learning. 

The design of simulation-based learning experiences in this study is informed by van Merrienboer’s and Kirschner’s instructional design model: Ten Steps to Complex Learning (van Merrienboer & Kirschner, 2017). This holistic instructional design model emphasizes that complex skills – like teaching – are characterized by the diversity of discrete elements in the task performance and the high degree of coordination required for high quality performance. At any given moment in a lesson, teachers are recalling key features of the lesson content, facilitating activities, monitoring the clock, and observing student work and behavior for evidence of comprehension. These simultaneous activities put a high demand on teacher cognitive load (Pass & van Merrienboer, 2020). For novices, it is challenging to both perform this entire complex assemblage and work on improving discrete parts at the same time. This observation is commonplace in many learning environments: it’s why music teachers and athletic coaches employ drills alongside recitals and scrimmages. In Ten Steps to Complex Learning, Kirschner & van Merrienboer refer to these drills or discrete practice opportunities as “part-task practice”, where novices can develop automaticity in managing recurrent features of the complex assemblage of the whole. Reich (2022) refers to part-task practice of discrete teaching elements as “teaching drills”. Since effective drilling involves high levels of repetition, digital simulations may play an important role in providing PSTs with sufficient practice time and repetition to improve in discrete elements that can be combined into the complex assemblage of teaching. 

R2.3 The discussion could be enlarged by explaining whether any similarities and discrepancies with other published data have been identified and accounted for. 

The discussion currently incorporates findings from four references: Levy (n.d.), Duckor (2016), Raymond et al (2013), and Self (2016). We have added references to Shaughnessy et al (2019), and Harlen (2015), and made more specific connections between the background section and the discussion section.   

Round 2

Reviewer 2 Report

Dear authors,

congratulations on improving your manuscript. You have significantly improved the clarity of your writing and have addressed most of my concerns.

Kind regards

The Reviewer